# Intelligent Clinical Decision Support

**DOI:** 10.3390/s22041408

**Published:** 2022-02-12

**Authors:** Michael R. Pinsky, Artur Dubrawski, Gilles Clermont

**Affiliations:** 1Department of Critical Care Medicine, University of Pittsburgh School of Medicine, Pittsburgh, PA 15261, USA; cler@pitt.edu; 2Auton Laboratory, School of Computer Science, Carnegie Mellon University, Pittsburgh, PA 15213, USA; awd@cs.cmu.edu

**Keywords:** database, machine learning, hemodynamic monitoring, predictive analytics

## Abstract

Early recognition of pathologic cardiorespiratory stress and forecasting cardiorespiratory decompensation in the critically ill is difficult even in highly monitored patients in the Intensive Care Unit (ICU). Instability can be intuitively defined as the overt manifestation of the failure of the host to adequately respond to cardiorespiratory stress. The enormous volume of patient data available in ICU environments, both of high-frequency numeric and waveform data accessible from bedside monitors, plus Electronic Health Record (EHR) data, presents a platform ripe for Artificial Intelligence (AI) approaches for the detection and forecasting of instability, and data-driven intelligent clinical decision support (CDS). Building unbiased, reliable, and usable AI-based systems across health care sites is rapidly becoming a high priority, specifically as these systems relate to diagnostics, forecasting, and bedside clinical decision support. The ICU environment is particularly well-positioned to demonstrate the value of AI in saving lives. The goal is to create AI models embedded in a real-time CDS for forecasting and mitigation of critical instability in ICU patients of sufficient readiness to be deployed at the bedside. Such a system must leverage multi-source patient data, machine learning, systems engineering, and human action expertise, the latter being key to successful CDS implementation in the clinical workflow and evaluation of bias. We present one approach to create an operationally relevant AI-based forecasting CDS system.

## 1. Introduction

There is a need for Artificial Intelligence (AI)-based tools in acute care environments to aid in the early detection and assessment of cardiorespiratory insufficiency (CRI) because the amount of information exceeds human capacity to process it, internalize the extracted knowledge, and then act upon it consistently and appropriately. The new onset of CRI is common in acutely ill hospitalized patients. Misdiagnosis and/or delayed treatment leads to increased morbidity, mortality, and cost of care [1]. Instability manifests through acute but frequently subtle changes in vital signs (VS) trends indicative of attempted compensation and evolving decompensation. Decompensation and instability occur even in highly monitored patient care environments such as in the Intensive Care Unit, and the longer a patient is in such a decompensated state the more difficult it is to mitigate or reverse resultant damage at the organ and cellular levels [2,3]. Being able to detect impending instability, rather than detecting its presence at a late stage, could permit timely stabilization, thereby reducing morbidity, mortality, and resource use. However, the technology needed to forecast impending instability is not well developed. Many early alerting approaches, once embedded in Electronic Health Record (EHR) systems, have subsequently been rolled back for a variety of reasons, including unacceptable performance, lack of perceived clinical usefulness, interference with existing workflows, and increased clinician documentation burden [4,5]. There is a clear need for better performing, trustworthy models that can be effectively melded into the clinical workflow. We and others have used machine learning (ML) to detect patterns predictive of impending instability before overt manifestations occur from real-time physiologic monitoring, often linked to EHR data. Such advanced intelligent clinical decision support (CDS) is only the first step towards developing AI-based systems that provide trustworthy, personalized predictions and recommendations to preemptively mitigate instability [6,7], hopefully leading to improved patient-centered outcomes.

Importantly, CRI usually develops over time and, therefore, it can potentially be predicted. Many researchers have demonstrated the feasibility of forecasting its overt onset. For instance, we have shown in animal models, step-down units, and ICUs that, typically, hemodynamic and respiratory instability develops over a time scale where clinical mitigations could be initiated in advance of the overt manifestations of instability [8,9,10,11,12,13,14,15,16,17,18,19,20]. However, creating AI tools with good performance is not enough. This perspective paper describes one approach to creating an operationally relevant AI-based forecasting CDS system.

## 2. Challenges of AI-Based CDS

AI-based tools must be deemed useful and trustworthy by end-users. There are two primary and very different challenges associated with using AI-based CDS tools at the bedside. The first is a lack of clinician trust in the system, a subject we and others have studied [21]. The other, less well-recognized challenge relates to the over-reliance on AI-based decision support [22,23,24]. CDS tools should be developed using sound human factor engineering principles to minimize information overload, to operate cohesively within the clinical workflow, and in support of an a priori expectation of incremental usefulness if validated.

Regrettably, AI-based tools may be biased. Bias in AI algorithms has been known to plague business applications for years. More recently, much emphasis has been put on exploring sources of biases in healthcare AI, including algorithms based on EHR [25,26,27,28,29]. Several landmark papers demonstrated that the naïve application of ML models to data may lead to erroneous conclusions or lack robustness across populations and subgroups of interest [30]. Statisticians have developed sophisticated methods to deal with observed and unobserved confounding, such as propensity-based methods [31,32]. Moreover, the issue of fairness in machine learning has recently become a very active research area, especially in healthcare applications [33]. Future predictive models should go beyond a simple examination of the impact of sex, race, and age as biological variables on model predictions. They should systematically explore data, algorithms, and results for the presence of bias. Potentially, models drawn from high-frequency data (i.e., 1 Hz and 250 Hz), when available, may be less exposed to certain forms of bias. However, this assumption needs to be validated across different clinical datasets to ensure their broad and consistent applicability [34].

Hence, AI-based tools need to be robust across environments. External validation of prediction algorithms and CDS tools is essential for the scalable impact of AI-based solutions. Creating generalizable tools is a difficult problem, for which there exist several non-mutually exclusive approaches. These approaches include: (1) external testing and transfer learning; (2) learning from multi-site broadly representative datasets; (3) federated approaches to learning. There are challenges and limitations to each of these methods. The current cybersecurity and privacy protection climate around health care data severely impedes the sharing of large datasets, including patient-level data. There is also a limit as to the notion of the generalizability of models. After all, predictions need to be accurate in a local environment, as trust is developed locally. When and how generally robust models can enhance local performance at sites where they were not developed remains to be explored [35].

Performing AI-based CDS in real time poses additional challenges. Although there are a growing number of tools for real-time decision support, most are rule-based and focused on detection (e.g., bedside alarms), rather than on personalized forecasting and therapeutics. There are some recent inspiring examples of promising work towards automated EHR-based early warning systems [36,37,38] and the forecasting of hypotension [39], which have impacted clinical workflow. However, current state of the art applications do not draw from multi-resolution, multi-domain data, including unstructured EHR data and monitor-derived waveform data. Furthermore, system architecture to build real-time AI-based systems needs to be not only developed locally, but also to maintain inter-institutional applicability. Such system requirements are not trivial to satisfy in practice.

## 3. Examples of Forecasting and Phenotyping Instability in the ICU

We have developed predictive models using controlled animal laboratory data and historical ICU data that demonstrate good performance in predicting clinically relevant tachycardia, hypotension, and bleeding. We also demonstrated the incremental benefit of high-frequency data in increasing the reliability of these prediction models [8,17,39,40,41]. For instance, advanced signal processing predicted clinically relevant tachycardia and hypotension in ICU patients [42,43]. To determine a clinically relevant tachycardia target in these studies, we first examined tachycardia’s impact upon outcomes from the MIMIC-III database. We found that although increasing HR > 100/min was associated with progressively increasing vasopressor use, morbidity, and mortality, a clear step-up in the length of stay and mortality occurred with HR > 130/min. Those tachycardia patients with a HR > 130/min had increased vasopressor support, longer ICU stays, and increased ICU mortality. Thus, we defined clinically relevant tachycardia as HR > 130/min, lasting ≥ 5 min with >10% density of occurrence over this time interval. Using data sampled at 1 Hz from 2809 subjects, classifiers were trained to create a risk score for future tachycardia [44]. Risk trajectory was generated from time windows moving away from the tachycardia event at 1-min increments. The classifiers performed generally well. The area under receiver operating characteristic curve (AUROC) score ranged from 0.842 when a regularized logistic regression model was used to 0.921 when a random forest (RF) classifier was used. Risk trajectory analysis showed average risks for the tachycardia group of 0.78 just before the tachycardia episodes, while control group risks remained <0.3 and with significant separation between subsequent tachycardia and control stable patients at ~75 min before the initial tachycardia event (Figure 1).

We also applied AI tools to predict hypotension, defined as systolic blood pressure < 90 mmHg and a mean arterial pressure <60 mmHg for ≥5 measurements within 10 min [13]. We used an RF classifier to predict hypotension and performance was measured by AUROC and the area under the precision-recall curve (AUPRC). We identified 1307 cases and matched them to 1619 non-hypotensive controls. The RF model showed AUROC of 0.93, 0.91, and 0.88 at 15, 30, and 60 min, respectively, before hypotension and AUPRC of 0.77 at 60 min before hypotension. Mean risk trajectory showed a clear separation from mean control risk trajectory >15 min before hypotension in 80% of cases. Since alerts predicting impending hypotension may also cause alarm fatigue if they are presented to the bedside clinicians too often, a second-level RF model analyzed the recent shape of the risk trajectory, combined with the existence of prior alerts, to generate potential alerts. We then imposed a lockout period of 15 min, where the system would not re-alert, even if alert conditions persisted. The resulting alert system produced on average 0.79 alerts/subject/hour, with a positive predictive value (probability of developing hypotension) of 65% and sensitivity of 92.4%. Thus, using this strategy to minimize alarm events and alarm fatigue did not materially impede the performance of this model.

We formally evaluated improvements in the performance of an instability model attainable when moving from non-invasive monitoring (NIM) to adding central venous, pulmonary artery, and arterial catheterization-derived variables and, as their sampling frequency was increased, from simple metrics (SM), computed every minute, to heart-beat-to-beat (B2B), to waveform (WF) with and without personalized stable baseline reference in our porcine 20 mL/min bleed cohort [41]. RF classification was used to identify the onset of bleeding. Model performance was evaluated using the AUROC curve, as well as the activity monitoring operating characteristic (AMOC) curve. The AMOC curve displays the tradeoffs between an earlier time to detection (i.e., how early bleeding can be detected after its onset) and increased false alarm rate (FPR). Referencing models to a personal stable baseline before a bleed improved bleed detection, as did an increase in data granularity from SM to B2B to WF. All invasive monitoring out-performed NIM (Figure 2). Thus, these data demonstrate that using more invasive monitoring-related data, increasing sampling frequency, and referencing to a personal baseline, cumulatively improve the detection of bleeding onset.

The limited availability of sufficiently large collections of clinically assessed reference data is one of the major bottlenecks preventing the wider development and adoption of robust AI-based CDS in practice. However, reviewing large amounts of clinical time series data to identify CRI can be both time-consuming and fraught with inter-rater variability of labeled events. To address these issues, we developed and applied an efficient protocol for a multi-expert, multi-tier ground truth elicitation framework with application to artifact classification for predicting patient instability [44], efficiently utilizing precious human expertise, and yielding accurate downstream models with one-quarter of the amount of time needed if conducted by the content experts manually. We also developed an active learning algorithm that prioritized which instances of data should be labeled by humans to maximally boost the eventual performance models, further reducing by half the need for direct visualization reviews while maintaining human interpretability of resulting model predictions [45,46]. In addition to being labor-intensive, the clinical expert data annotation process is often prone to error and uncertainty, especially if the cases are to be assessed individually. Our studies showed that more reliable labels of real versus artifact, or minor versus clinically relevant instability, can be collected by asking other kinds of questions. For instance, supplementing direct labeling of each data instance with qualitative comparisons such as: “Comparing patient A and B, who appears healthier?” [47,48]. Answering such questions is often easier, yields more reliable labels, and requires less annotation effort to achieve equivalent performance. We have also demonstrated how to completely avoid the laborious process of pointillistic labeling of reference data for clinical applications of AI using weak supervision [49]. By harvesting multiple labeling functions that a human expert would use in their mind to assess each case at hand, one can automate the process of data annotation. This is particularly appealing when facing large amounts of clinical data needing annotation. In one such exercise, we have shown that a handful of labeling functions derived from basic clinical knowledge can eliminate the need for manual data annotation and yield predictive models of performance comparable to the equivalent models trained on data point-by-point labeled by expert clinicians when evaluated on the task of detecting arrhythmia in ECG signals [49].

Finally, the importance of a usable and informative human–computer interface contributes to user trust in a major way and is often overlooked. Unless a CDS works autonomously, it requires a machine–human interface to allow the clinicians to both operationalize the alerts and corresponding information and to audit its performance. Traditionally, such interfaces included a visual display from an electronic monitor, either a bedside monitor or handheld device such as a tablet or smartphone. We and others have proactively involved the end-users in graphical user interface (GUI) development [50]. We demonstrated that clinical end-users are usually eager to engage in GUI development and readily provide useful feedback on issues related to both the GUI design and the foundational CDS architecture. This feedback revealed specific items that bedside clinician users found important, including a better trend evolution display and context of alert relative to overall care. Unfortunately, bypassing input from clinical experts during the early-stage development of a GUI CDS system is quite common [51,52].

## 4. Conclusions

AI-based CDS processes need to be highly iterative using multiple pathways and forms of feedback involving both the modelers and the target clinicians, linked by in silico trials of effectiveness and acceptance by end-users and using clear metrics of success. One can never fully eliminate bias, but newer AI approaches may be able to mitigate several of the sources of bias CDS systems contend with, yielding diagnostic, predictive, or prescriptive tools that optimize accuracy and preserve the fairness of the recommended decisions across subpopulations. Any AI-based CDS will have a finite lifecycle and will require periodic reevaluation and refinement to adapt to changing patient demographics, data ecosystems, bedside workflows, and evolution in clinical practices if they are to sustain effectiveness and trustworthiness. Some of these needs can be automated or semi-automated with the use of AI. For instance, efficient methods for acquiring training information for models will likely be able to facilitate such adaptivity at operationally feasible costs. End-user involvement in the design and evaluation of CDS systems across their lifecycle will also promote trustworthiness, adoption into practice, and sustainability.

Future research needs to optimally leverage multi-institutional datasets to not only develop clinically relevant predictive models, but also to establish effective and efficient methodologies to plumb these databases within the context of health information security and democratization. Such broad-based efforts are underway [53] and, hopefully, will lead to novel and insightful ways of joining models and applications across groups of investigators and across healthcare facilities, once developed.

## Figures and Tables

**Figure 1 sensors-22-01408-f001:**
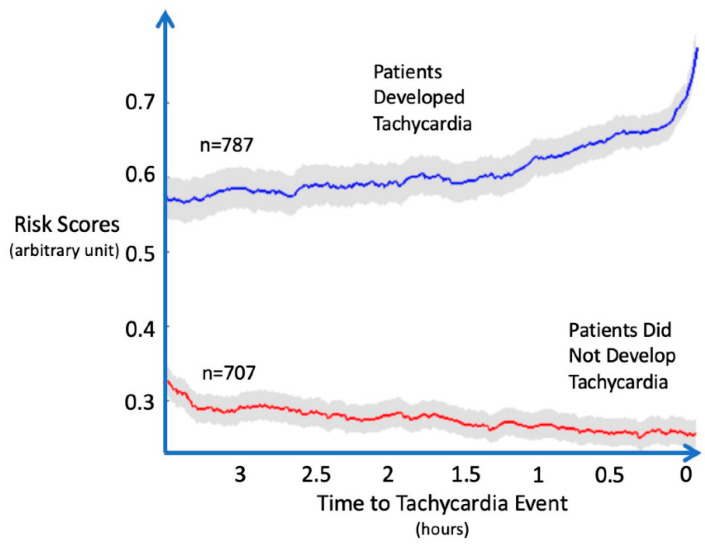
Model prediction of initial tachycardic episode using external control data matched for every episode of tachycardia. Comparison of a model trained on MIMIC-II data to identify an initial episode of tachycardia (heart rate (HR) > 130/min) in an external validation cohort from that same database. Results are shown as risk score changes over time as the future tachycardic and non-future tachycardic (control) groups move toward the event. The control group’s time series data were time-matched to correspond to the future tachycardic group’s time in the ICU. Data derived from Yoon et al. [43].

**Figure 2 sensors-22-01408-f002:**
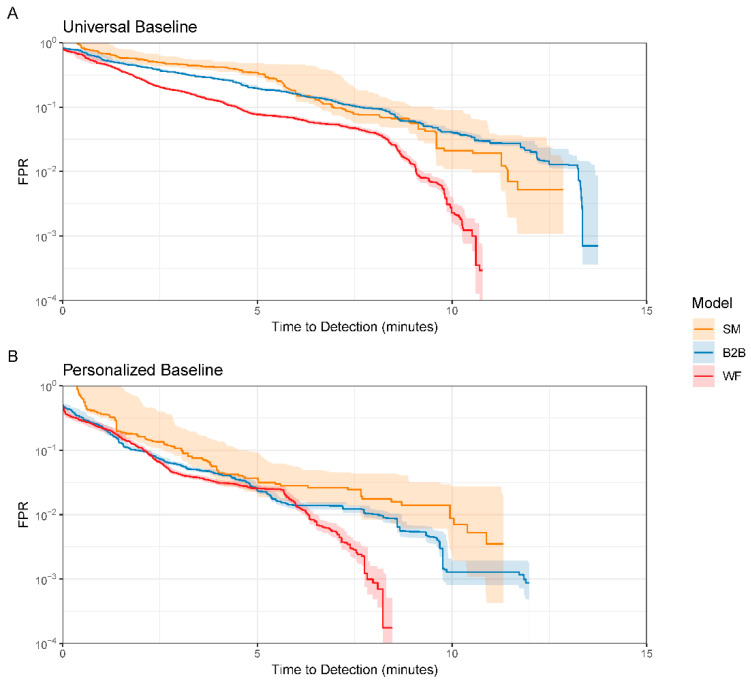
Activity monitoring operating characteristic analysis of models developed with increasingly granular arterial pressure physiologic data, for models developed using a universal baseline (**A**), and models developed using a personalized baseline (**B**) (see text for details). Displayed as the time to detection of bleeding versus false-positive rate for arterial catheter data only for increasing granularity levels: simple metrics (SM), beat-to-beat (B2B), and waveform (WF). Data displayed with shading equal to 95% confidence range. Data derived from Pinsky et al. [41].

## Data Availability

Data are available upon request. Contact pinskyme@pitt.edu to initiate such a request.

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
