# Peer review of "Intelligent Clinical Decision Support"

_sensors, 2022, doi:10.3390/s22041408_

Round 1

Reviewer 1 Report

The authors present very interesting research and proposals to support clinical decisions using AI models. Although the topic is interesting and the conclusions presented in the article are promising, it is quite difficult to read this work. I understand that this is not a mdpi article but a mdpi perspective and maybe that is why, according to the editor's wish, it should look like this, but I think that if the research results were to be described in accordance with the rules of art, refer to the literature so richly presented at the end and not referred to in the text, it would be a good article to support a more detailed discussion. So if this is the definition of perspective from mdpi then ok. Publishable. If I were to judge it in the direction of the article, there would be much room for improvement, although the research and results are promising.

Author Response

We thank this reviewer for their insightful and specific comments. Yu are correct, this manuscript was an invited Perspective piece, not an original research submission. We also spent a fair amount of time copy editing the final version of this manuscript to make it read more clearly. 

Reviewer 2 Report

This is a generally well-written and structured paper with sufficient theoretical background. In this paper, the authors present Intelligent Clinical Decision Support. In my opinion, it could be published if the following issues are resolved.

1.The writing of the paper needs a lot of improvement in terms of grammar, spelling, and presentations. The paper needs careful English polishing since there are many typos and poorly written sentences.

Some examples are as the following:

*     Check the usage of the commas carefully.

*     Check the articles including "a", "an" and "the".

*     Check the required and unneeded blank spaces.

2. Please, motivate more the abstract, trying to be more concise. Why this work is necessary?

3. The abstract does not reflect contribution of the study.

4. Avoid repetitions. I can see several repetitions at different places in this paper. A thorough proofreading is required.

5. intensive care unit (ICU), artificial intelligence (AI), and clinical decision support (CDS), please change to the Intensive Care Unit (ICU), Artificial Intelligence (AI), and Clinical Decision Support (CDS)  in the Abstract.

6. The Intensive Care Unit (ICU) environment is particularly well-positioned to demonstrate the value of AI in saving lives. Authors should write once a time this words Intensive Care Unit (ICU) in the Abstract. After that, should use the abbreviation.

7. In the Introduction, 

*vital signs (VS)=Vital Signs (VS)

*electronic health record systems (EHR)=Electronic Health Record systems (EHR)

*machine learning (ML)=Machine Learning (ML)

8. There is no scope for future research. A clear direction for future research is required.

In my opinion, I hope my comments are useful. So I suggest a revision. I look forward to receiving the revised version in due time.

Author Response

We thank this reviewer for their insightful and specific comments. You are correct, this manuscript was an invited Perspective piece, not an original research submission. We also spent a fair amount of time copy editing the final version of this manuscript to make it read more clearly.  The first author is a native English speaker, so this task was relatively straightforward. Also, the specific formatting issues and use of “a” and “the” were reviewed throughout the text to make them read more clearly as suggested this this reviewer.